# Fatty Pancreas and Cardiometabolic Risk: Response of Ectopic Fat to Lifestyle and Surgical Interventions

**DOI:** 10.3390/nu14224873

**Published:** 2022-11-17

**Authors:** Kok Hong Leiu, Sally D. Poppitt, Jennifer L. Miles-Chan, Ivana R. Sequeira

**Affiliations:** 1Human Nutrition Unit, School of Biological Sciences, University of Auckland, Auckland 1024, New Zealand; 2High Value Nutrition, National Science Challenge, Auckland 1010, New Zealand; 3Department of Medicine, University of Auckland, Auckland 1010, New Zealand; 4Riddet Centre of Research Excellence (CoRE) for Food and Nutrition, Palmerston North 4442, New Zealand

**Keywords:** pancreatic fat, liver fat, cardiometabolic health, energy restriction

## Abstract

Ectopic fat accumulation in non-adipose organs, such as the pancreas and liver, is associated with an increased risk of cardiometabolic disease. While clinical trials have focused on interventions to decrease body weight and liver fat, ameliorating pancreatic fat can be crucial but successful intervention strategies are not yet defined. We identified twenty-two published studies which quantified pancreatic fat during dietary, physical activity, and/or bariatric surgery interventions targeted at body weight and adipose mass loss alongside their subsequent effect on metabolic outcomes. Thirteen studies reported a significant decrease in body weight, utilising weight-loss diets (*n* = 2), very low-energy diets (VLED) (*n* = 2), isocaloric diets (*n* = 1), a combination of diet and physical activity (*n* = 2), and bariatric surgery (*n* = 5) including a comparison with VLED (*n* = 1). Surgical intervention achieved the largest decrease in pancreatic fat (range: −18.2% to −67.2%) vs. a combination of weight-loss diets, isocaloric diets, and/or VLED (range: −10.2% to −42.3%) vs. diet and physical activity combined (range: −0.6% to −3.9%), with a concurrent decrease in metabolic outcomes. While surgical intervention purportedly is the most effective strategy to decrease pancreas fat content and improve cardiometabolic health, the procedure is invasive and may not be accessible to most individuals. Given that dietary intervention is the cornerstone for the prevention of adverse metabolic health, the alternative approaches appear to be the use of weight-loss diets or VLED meal replacements, which are shown to decrease pancreatic fat and associated cardiometabolic risk.

## 1. Introduction

Ectopic fat infiltration in non-adipose organs, such as muscles, heart, liver, and pancreas, is detrimental to the regulation of cardiometabolic health [1]. This infiltration is commonly attributed to positive energy balance and, in turn, is associated with body weight and adipose mass gain. The ectopic fat infiltration occurs due to ‘impaired’ subcutaneous adipose tissue (SAT) storage, leading to lipid ‘*overspill*’ into secondary deep subcutaneous visceral adipose tissue (VAT) compartments and ectopic sites [2] also termed lipotoxicity, or alternatively, it is hypothesised to be due to impaired adipogenesis or limited adipocyte expandability [3]. As a result, the excess accumulation of ectopic fat with the presence of insulin resistance promotes the development of lipotoxicity, which might lead to multiple organ dysfunction or even failure [4].

The recently coined term ‘thin on the outside, fat on the inside’ (TOFI) has been used as a descriptor for individuals with little subcutaneous peripheral or abdominal fat, yet high levels of ectopic or intra-organ fat [5,6]. The increased lipotoxic milieu has been associated with, as well as predictive of, the development of insulin resistance en route to type 2 diabetes (T2D) in susceptible individuals [1]. The accumulation of ectopic fat in different organs has been linked with several cardiometabolic risks, such as atherosclerosis, cardiomyopathy, and renal dysfunction [7,8,9]. This emerging association between ectopic fat accumulation and cardiometabolic risk has been supported by a recent position statement suggesting that VAT is an independent risk factor of cardiometabolic morbidity and mortality [10].

While the relationships between the risk factors that underpin the deposition of ectopic fat in these key metabolic organs remain unknown, they may include ethnicity [11,12], gender [13], diet [14,15], and physical activity [16,17]. In principle, attenuating risk via the development of intervention strategies targeted at decreasing ectopic fat stores may provide a highly successful approach to ameliorate cardiometabolic risk. While clinical trials have focused on interventions to decrease body weight and liver fat, there is an emerging role for concurrently decreasing fatty pancreas given its significant role in regulating glucose metabolism. Notably, the pancreas in rodent [18] and human [19] studies have been shown to have a five-fold greater susceptibility to lipid infiltration in comparison to the liver. Hence, the current narrative review presents a summary of intervention trials conducted to date that report changes in ectopic fat stores, with a particular focus on pancreatic fat; during the process of dieting, exercise or surgical procedures targeted at body weight and adipose mass loss and established metabolic biomarkers. With recent advancements and the standardisation of non-invasive ectopic scanning technologies, there has been a renewed focus on ameliorating fatty pancreas across a range of population groups; however, successful intervention strategies remain to be explored.

## 2. Pancreatic Fat—Are We Overlooking a Key Piece of the Metabolic Puzzle?

While numerous intervention studies have quantified changes in liver fat under various conditions, [20,21,22,23,24,25,26] few have yet to focus on pancreatic fat. One of the main reasons for this has been the absence of robust imaging techniques as the retroperitoneal position of the pancreas makes the scanning of the organ challenging compared to other organs, such as the liver [27]. Recent advancements in imaging techniques enable the location of the pancreas and quantification of both lipid infiltration and losses from the organ across a range of intervention protocols. The pancreas is located posterior to the upper abdomen, behind the stomach, and consists of three regional compartments, i.e., the head, body, and tail. The former is surrounded by the duodenum while the latter extends to the hilum of the spleen, with the body lying posterior to the pyloric region of the stomach. It is reported that there is a greater tendency for the deposition of fat in the body and tail than in the head of the pancreas, particularly in individuals who are at risk of T2D [28].

Moreover, several morphologic studies have reported variability in pancreas volume due to factors such as age, gender, ethnicity, and comorbidities. T2D has been shown to be associated with a smaller pancreas volume [29], with the length and duration of diabetes status further associated with decreased volume [30] due to parenchymal atrophy resulting from either dysfunction/death of pancreatic islet β-cells or impaired function of the insulin receptor on pancreatic acinar cells [31]. Pancreas volume is also reportedly larger in adult males (~86 cm^3^) than females (~73 cm^3^) [13] and commonly decreases after 60 years of age [29]. Individuals of Asian descent have been reported to have smaller pancreas volume compared to Caucasians, when matched for both body mass index (BMI) and body fat mass [11], possibly due to smaller body size/stature. Furthermore, BMI is shown to be positively correlated with pancreas volume among Asians [12], despite the propensity for the TOFI phenotype and predicted higher pancreatic fat in Asian individuals with low total body fat.

Apart from its structure, the pancreas also has both exocrine and endocrine functions that significantly contribute to metabolic health [32]. The former is a tubular network made of acinar and duct cells comprising more than 95% of pancreatic parenchyma that produces, secretes, and transports digestive enzymes into the duodenum during digestion, whilst the latter comprises only about 2% of parenchyma that produces and secretes peptide hormones, including insulin and glucagon into the blood to regulate glucose homeostasis [32]. Therefore, pancreatic function may play an early and vital role in the development of T2D among people with overweight and obesity, including those with the TOFI phenotype [33,34,35]. In such individuals, it is purported that increased intra-pancreatic fat acutely impairs insulin secretory responses following a meal, to perpetuate the longer-term development of hyperglycaemia and T2D.

## 3. Adverse Metabolic Effects Attributed to Accumulation of Pancreatic Fat

Intra-pancreatic fat deposition (IPFD), which has been defined as the diffuse presence of fat within the pancreas [36], is posited to occur through two major pathways [37]. Firstly, adipocytes may infiltrate the pancreatic parenchyma, with triglyceride (TAG) stored as large central lipid droplets within the adipocyte. Secondly, lipid droplets may form directly within non-adipocyte cells of the pancreas [36]. Despite there being little evidence to underpin the mechanisms directing these outcomes, it is still possible to morphologically characterise IPFD. The pancreas comprises multiple lobules, with fat deposition occurring both within lobules (intra-lobular fat) and between lobules (inter-lobular fat) [37]. The intra-lobular fat includes both lipid droplets within endocrine and acinar cells, and adipocytes from the trans-differentiation of acinar cells and the replacement of apoptotic acinar cells. Conversely, inter-lobular fat comprises TAG-rich adipocytes between the lobules, as well as lipid droplets released by the activated pancreatic stellate cells in response to damage to the pancreas. As a result of this infiltration by both adipose cells and lipid droplets, excessive IPFD in the long term can lead to the development of pancreatic steatosis.

A growing body of literature implicates pancreatic steatosis and lipotoxicity with dysfunction or apoptosis of pancreatic cells [38]. Lipotoxicity impairs pancreatic function via a number of mechanisms, including endoplasmic reticulum (ER) stress, oxidative stress, mitochondrial dysfunction, islet inflammation, and beta-cell apoptosis [39]. Firstly, ER stress has been shown to occur due to cellular demands from the ER to facilitate the increased production of hormones, such as insulin and glucagon to maintain glucose homeostasis. As a result, ER stress triggers the unfolded protein response (UPR) to maintain homeostasis within the ER, with cellular apoptosis shown to occur if UPR is unable to relieve ER stress [40]. Secondly, the overproduction of the reactive oxygen species (ROS) during oxidative stress decreases the production of adenosine triphosphate (ATP) by the mitochondria due to mitochondrial DNA and membrane protein damage [41]. Thirdly, circulating free fatty acids (FFAs) activate genes associated with inflammation and the expression of chemokines to trigger an immune response within the pancreatic beta-cell [42]. Lastly, FFAs also induce beta-cell apoptosis via saturated FFAs (e.g., palmitate), which activate the nuclear factor (NF)-kB and upregulate inducible nitric oxide synthase (iNOS) to induce apoptosis [43,44].

The dysfunction or apoptosis of pancreatic cells directly affects pancreas function, in turn disrupting blood glucose homeostasis, including enzymatic production involved in digestive processes. The accumulation of fat in the pancreas is reported to be associated with a number of adverse clinical consequences, including impaired glucose metabolism, decreased insulin sensitivity, metabolic syndrome and acute pancreatitis [45]. A prior study in Asian Chinese reported more cases of metabolic syndrome-related outcomes, including abdominal obesity, hyperglycaemia, hypertension and hypercholesterolaemia among individuals with worsening of pancreatic fat, i.e., steatosis [46]. Moreover, these individuals exhibited hyperinsulinaemia and impaired insulin sensitivity [47,48] underpinning impaired glucose metabolism.

Despite this reported association with T2D risk factors, there is still no strong evidence to support a causal relationship between pancreatic steatosis and the development of frank T2D [49,50,51,52]. A recent longitudinal study in a Japanese cohort of ~200 individuals without diabetes did not report an association between pancreatic steatosis and risk of T2D [53,54], and hypothesised that pancreatic fat may not be a cause but rather a manifestation of dysglycaemia. However, the authors have since reported that lipid accumulation in the pancreas did increase the risk of developing T2D, and notably, was independent of BMI in a cohort of nearly 1500 non-diabetic Japanese with BMI within the lean range [55] and were typified by the TOFI phenotype. Longitudinal follow-up data from this Asian cohort, with a 4% incidence of T2D over 6 years, importantly led the authors to conclude that T2D was more likely to develop in individuals with BMI within the lean range but with the fatty pancreas, and that in a cohort with neither obesity nor overweight, pancreatic steatosis can be used to identify those at high risk for the later development of T2D.

Preclinical studies contribute surprisingly little to the literature, with few investigating causal links between IPFD and T2D. Obese mice fed a high-fat diet accumulated significant TAG within the pancreas which, in turn, led to early onset T2D compared to normal-weight mice [56,57]. Metabolic disorders (e.g., hyperinsulinemia, dyslipidaemia and hyperglycaemia) have also been shown to be resultant from ectopic fat accumulation [58].

Evidence regarding associations between pancreatic steatosis and frank T2D [29,30] is also incongruent. Lim and colleagues [30] reported that individuals >5 years post-diagnosis of T2D had the highest pancreatic fat content, followed in turn by those <5 years post-diagnosis, newly diagnosed T2D, and absent of T2D. Conversely, Saisho and colleagues [29] found that computed tomography (CT) assessed that pancreatic fat content increased with age and BMI but did not differ between those with and without T2D. It is important to highlight that those studies measuring pancreatic fat content (PFC) have utilised a range of assessment methods. For instance, CT is unable to differentiate between fat stored in adipocytes or parenchymal cells, which may contribute to the underestimation of PFC. The techniques used to assess ectopic organ fat may bias the quantification of fat depots, with different scanning techniques such as ultrasound, CT, and MRI, each having advantages and disadvantages [2,59], as presented in Table 1. Whilst all these scanning methods are non-invasive, the images obtained between studies will have likely been generated in different formats and using variable softwares for analysis.

Despite these limitations, evidence underpinning the association between pancreatic fat and cardiometabolic endpoints is growing whereby pancreatic fat has been reported to be significantly correlated with several cardiometabolic endpoints, such as TAG and high-density lipoprotein cholesterol (HDL-C) [60]. Recent publication have also reported that the presence of pancreatic fat in the body increases the risk of developing metabolic comorbidities (e.g., metabolic syndrome, insulin resistance, non-alcoholic fatty liver disease (NAFLD), hypertension and atherosclerosis) [8,61]. The narrative review presented below provides a summary of interventions that have assessed changes in ectopic fat stores during dietary, exercise or surgical procedures targeting body weight and adipose mass loss, with a particular focus on pancreatic fat.

## 4. Clinical Intervention Strategies—Targeting Amelioration of Ectopic Fat

There have been a number of clinical interventions that have utilised a combination of diet, physical activity, and/or bariatric surgery to determine the effects on body weight, body fat compartments, and metabolic parameters. Table 2 presents a summary of these lifestyle and surgical intervention trials reporting effects on pancreatic fat, alongside other key parameters related to metabolic outcomes. These comprise both (i) multi-arm randomised controlled trials (RCTs) and (ii) single-arm interventions with no comparator treatment group, where outcomes have been presented as within-treatment change from baseline.

### 4.1. Weight-Loss Diet Interventions

Three studies have utilised a weight-loss dietary regime to determine the impact on body fat compartments including pancreatic fat and metabolic parameters. The first study [62] utilised a single-arm 500 kcal dietary restriction below resting energy expenditure (REE) over a 6-month duration among individuals with obesity. Notably, no comparator or control arm was assessed in this trial, limiting the interpretation of the outcomes. However, a significant 9% decrease in body weight among individuals with obesity resulted in a parallel decrease in VAT (−31.9%), PFC (−42.3%) and liver fat content (LFC) (−84.1%). In addition, most of the metabolic parameters assessed also significantly improved, except for HDL-C.

A second study, the HELENA Trial, was a well-designed 3-arm RCT [15] which assigned individuals with overweight and obesity into different energy restriction groups over 50 weeks. Both intermittent calorie restriction (ICR) and continuous calorie restriction (CCR) significantly decreased body weight (ICR: −4.6% vs. control: −1.0%; CCR: −5.0% vs. control: −1.0%, *p* < 0.01) and VAT (ICR: −16.7% vs. control: −2.0%; CCR: −10.2% vs. control: −2.0%, *p* < 0.05) compared to the healthy balanced diet arm (control). In turn, CCR also significantly decreased LFC by ~20% compared to a mere ~5% in control (*p* < 0.01), but unexpectedly this effect was not observed with ICR despite similar decreases in body weight and VAT. Despite the positive outcomes in both ICR and CCR groups, there was no significant decrease in PFC or the metabolic parameters when assessed between the three groups at the end of the long intervention. The lack of significant outcomes was possibly due to the modest 5% weight loss in the CCR and ICR groups.

A third, shorter duration RCT [14] recently conducted in individuals with T2D utilised a different approach, by evaluating two diets in a 12-week cross-over design: a carbohydrate-reduced high protein (CRHP) vs. a conventional diabetes (CD) prevention diet. After 6 weeks of intervention, there was no significant difference in body weight loss (CRHP: −1.6% vs. CD: −0.9%) or VAT (CRHP: −5.6% vs. CD: −2.5%), not unexpected in such a short-duration intervention. Conversely, and unexpectedly considering the absence of body weight and VAT loss, the CRHP diet resulted in a significantly greater decrease in both PFC (CRHP: −27.9% vs. CD: −11.1%, *p* < 0.05) and LFC (CRHP: −41.4% vs. CD: +6.1%, *p* < 0.01) compared to the CD diet. In addition, the CRHP diet showed significant improvements in fasting plasma glucose (FPG) (CRHP: −7.8% vs. CD: −1.1%, *p* < 0.05), HbA_1c_ (CRHP: −10.8% vs. CD: −1.4%, *p* < 0.001) and TAG (CRHP: −26.7% vs. CD: 7.7%, *p* < 0.001) compared to the CD diet.

### 4.2. Very Low-Energy Diet (VLED) Total Meal Replacement Interventions

VLED is a specific dietary regime that utilises a meal replacement strategy, commonly using commercial liquid-based products, such as shakes, porridge and soups that provides less than 4 MJ/day. These dietary regimes are designed to promote rapid weight loss within 12 weeks while minimising muscle mass loss due to the high-protein content provided by the meal [63]. There have been three trials conducted using total meal replacement VLEDs (<4 MJ/day) and reporting pancreatic fat outcomes, all of which were single-arm interventions analysed as changes from the pre-intervention baseline. VLEDs result in a significant negative energy balance, which has consistently been shown to drive rapid acute weight loss and aid in T2D prevention and treatment [64,65,66,67]. Individuals with overweight and obesity and diagnosed T2D recruited into a study [68] had ~15% significant decrease in body weight over 8 weeks, as well as the corresponding ~23% decrease in PFC and ~77% decrease in LFC. There were also significant improvements in several metabolic parameters, including FPG, fasting insulin, HbA1c, and TAG, following the 8-week intervention period.

Another study utilising a similar 8-week VLED intervention among individuals with T2D, followed by an additional 6-month follow-up period [69], characterised their participants at the end of the study as responders (FPG < 7 mmol/L) or non-responders. They reported a significant decrease in body weight for both responders (−15.8%, *p* < 0.001) and non-responders (−13.6%, *p* < 0.001) after 8 weeks, followed by a minor rebound or weight regain for both responders (+0.4%, *p* > 0.05) and non-responders (+1.4%, *p* > 0.05) at the 6-month follow-up. However, there was no significant difference in the changes between the two groups (*p* > 0.05).

**Table 2 nutrients-14-04873-t002:** Various lifestyle and surgical trials reporting effects on pancreas fat and other key parameters related to metabolic outcomes.

Author (Year)	Duration, Population	Intervention	Imaging Technique	Effect on Body Weight	Effect on Body Fat Compartments	Effect on Metabolic Parameters Associated with Glycaemia
Dietary Interventions						
(a) Weight-loss diet						
Rossi et al. (2012) [62]	6 mo n = 24 (13 M, 11 F) 26 to 69 y non-T2D OB (BMI: 30 to 50 kg/m^2^)	Single arm (non-RCT) with hypo-energetic diet 500 kcal below REE × PAL of 1.4 diet composition 62 en% CHO, 24 en% FAT, 14 en% PRO, 20 g fibre plus water only	1.5 Tesla MRI (Symphony, Siemens, Erlangen, Germany)	Decrease at 6 mo −8.9%, *p* < 0.001	Decrease at 6 mo: VAT (−31.9%, *p* < 0.001) PFC (−42.3%, *p* < 0.01) LFC (−84.1%, *p* < 0.001)	Decrease at 6 mo for: FPG (−0.4%, *p* < 0.05) Insulin (−39.1%, *p* < 0.001) HOMA IR (−40.9%, *p* < 0.001) TAG (−27.0%, *p* < 0.05) ALT (−23.6%, *p* < 0.001) NS decrease at 6 mo. HDL-C (*p* > 0.05)
HELENA Trial Schubel et al. (2018) [15]	50 wk n = 150 (75 M, 75 F) 35 to 65 y non-T2D OW and OB (BMI: ≥25 to <40 kg/m^2^) 16% MetS	Three arms (RCT): ICR group 25% of required energy intake for 2 non-consecutive days and healthy balanced diet for remaining 5 days of the wk CCR group 80% of required energy intake daily Control group: healthy balanced diet	1.5 Tesla MRI (MAGNETOM Aera, Siemens, Erlangen, Germany)	Decrease at 50 wk −4.6% for ICR vs. −1.0% for control (*p* < 0.01) −5.0% for CCR vs. −1.0% for control (*p* < 0.01) No difference between ICR and CCR (*p* > 0.05)	Decrease at 50 wk:VAT −16.7% for ICR vs. −2.0% for control (*p* < 0.01) −10.2% for CCR vs. −2.0% for control (*p* < 0.05) No difference between ICR and CCR (*p* > 0.05) PFC No difference between ICR vs. CCR (*p* > 0.05) ICR vs. control (*p* > 0.05) CCR vs. control (*p* > 0.05) LFC −20.2% for CCR vs. −5.6% for control (*p* < 0.01) No difference between ICR and CCR (*p* > 0.05) ICR and control (*p* > 0.05)	NS decrease at 50 wk for ICR vs. control, CCR vs. control or ICR vs. CCR: FPG (*p* > 0.05) Insulin (*p* > 0.05) HOMA IR (*p* > 0.05) TAG (*p* > 0.05) HDL-C (*p* > 0.05) AST (*p* > 0.05) ALT (*p* > 0.05)
Skytte et al. (2019) [14]	12 wk n = 28 (20 M, 8 F) >18 y T2D OW and OB (BMI: ≥25 kg/m^2^)	Two arms (RCT), then cross-over after 6 wk: CRHP group 30 en% CHO, 30 en% PRO, 40 en% FAT CD group 50 en% CHO, 17 en% PRO, 33 en% FAT	3.0 Tesla MRI (Ingenia, Philips, the Netherlands)	Decrease at 6 wk No difference between CRHP and CD (*p* > 0.05)	Decrease at 6 wk: VAT No difference between CRHP and CD (*p* > 0.05) PFC −27.9% for CRHP vs. −11.1% for CD (*p* < 0.05) LFC −41.4% for CRHP vs. +6.1% for CD (*p* < 0.01)	Decrease at 6 wk for: FPG −7.8% for CRHP vs. −1.1% for CD (*p* < 0.05) HbA_1c_ −10.8% for CRHP vs. −1.4% for CD (*p* < 0.001) TAG −26.7% for CRHP vs. −7.7% for CD (*p* < 0.001) NS decrease and difference between CRHP and CD at 6 wk: HOMA IR (*p* > 0.05) HDL-C (*p* > 0.05)
(b) Low-energy diet (LED) total meal replacement						
Lim et al. (2011) [68]	8 wk n = 11 (9 M, 2 F) 35 to 65 y T2D < 4 y OW and OB (BMI: 25 to 45 kg/m^2^)	Single arm (non-RCT) with VLED Liquid diet 46 en% CHO, 33 en% PRO, 20 en% FAT, 510 kcal	3.0 Tesla MRI (Achieva, Philips, the Netherlands)	Decrease at 8 wk −14.8%, *p* < 0.05	Decrease at 8 wk: PFC (−22.5%, *p* < 0.05) LFC (−77.3%, *p* < 0.01)	Decrease at 8 wk: FPG (−41.3%, *p* < 0.01) HbA_1c_ (−26.3%, *p* < 0.01) Insulin (−57.0%, *p* < 0.05) TAG (−45.8%, *p* < 0.05) NS decrease at 8 wk HDL-C (*p* > 0.05) ALT (*p* > 0.05)
Steven et al. (2016) [69]	8 wk of VLED and follow-up until 6 mo n = 30 25 to 80 y duration of T2D (short: <4 y or long: >8 y) obese (BMI: 27 to 45 kg/m^2^) Analysed as responder (FPG < 7.0 mmol/L) and non-responder (FPG > 7.0 mmol/L) at end of trial	Single arm (non-RCT) with VLED Liquid diet 43 en% CHO, 34 en% PRO, 19.5 en% FAT, 624 kcal	3.0 Tesla MRI (Achieva, Philips, the Netherlands) using three-point Dixon method	Decrease at 8 wk −15.8% for responder ^a^ vs. −13.6% for non-responder ^b^ (*p* > 0.05) Increase at 6 mo +0.4% for responder ^a^ vs. +1.4% for non-responder ^b^ (*p* > 0.05)	Decrease at 8 wk: VAT −33.1% for responder ^a^ (*p* < 0.05); −27.7% for non-responder ^b^ (*p* < 0.05) PFC −17.0% for responder ^a^ (*p* < 0.05); −10.2% for non-responder ^b^ (*p* < 0.01) LFC −82.8% for responder ^a^ (*p* < 0.01); −73.2% for non-responder ^b^ (*p* < 0.001) Decrease at 6 mo: VAT −37.5% for responder ^a^ (*p* < 0.05); −31.3% for non-responder ^b^ (*p* < 0.05) PFC NS decrease for responder ^a^ and non-responder ^b^ (*p* > 0.05) LFC NS decrease for responder ^a^ and non-responder ^b^ (*p* > 0.05)	Decrease at 8 wk: Insulin −61.3% for responder ^a^ (*p* < 0.05); −40.9% for non-responder ^b^ (*p* < 0.05) TAG −35.0% for responder ^a^ (*p* < 0.05); −23.1% for non-responder ^b^ (*p* < 0.05) HDL-C NS decrease for responder ^a^ (*p* > 0.05); NS decrease for non-responder ^b^ (*p* > 0.05) ALT −39.5% for responder ^a^ (*p* < 0.05); NS decrease for non-responder ^b^ (*p* > 0.05) Decrease at 6 mo: Insulin −62.7% for responder ^a^ (*p* < 0.05); −36.6% for non-responder ^b^ (*p* < 0.05) TAG −40.0% for responder ^a^ (*p* < 0.05); −7.7% for non-responder ^b^ (*p* < 0.05) HDL-C 27.3% increase for responder ^a^ (*p* < 0.05); NS increase for non-responder ^b^ (*p* > 0.05) ALT −51.2% for responder ^a^ (*p* < 0.05); −4.0% decrease for non-responder ^b^ (*p* < 0.05)
DiRECT-Trial Al-Mrabeh et al. (2020) [70]	5 mo of VLED and follow-up until 24 mo n = 33 (19 M, 14 F) 20 to 65 y T2D < 6 y OW and OB (BMI: 27 to 45 kg/m^2^) Analysed as responder (FPG < 7.0 mmol/L) and non-responder (FPG > 7.0 mmol/L) at end of trial	Single arm (non-RCT) with VLED Liquid diet (825–853 kcal) from the Counterweight-Plus weight management program	3.0 Tesla MRI (Achieva, Philips, Netherlands) using a 3-point Dixon method, with gradient-echo scans	Decrease at 5 mo −16.1% for responder ^c^ (*p* < 0.001) −13.1% for non-responder ^d^ (*p* < 0.001) Decrease at 24 mo −11.5% for responder ^c^ (*p* < 0.001) −11.6% for non-responder ^d^ (*p* < 0.01)	Decrease at 5 mo: VAT −43.0% for responder ^c^ (*p* < 0.001); −35.9% for non-responder ^d^ (*p* < 0.001) PFC −10.3% for responder ^c^ (*p* < 0.001); −10.1% for non-responder ^d^ (*p* < 0.01) LFC −79.6% for responder ^c^ (*p* < 0.001); −82.1% for non-responder ^d^ (*p* < 0.001) Decrease at 24 mo: VAT −27.2% for responder ^c^ (*p* < 0.001); −27.5% for non-responder ^d^ (*p* < 0.05) PFC −8.0% for responder ^c^ (*p* < 0.001); NS decrease for non-responder ^d^ (*p* > 0.05) LFC −60.5% for responder ^c^ (*p* < 0.001); NS decrease for non-responder ^d^ (*p* > 0.05)	Decrease at 24 mo: FPG −33.3% for responder ^c^ (*p* < 0.001); NS decrease for non-responder ^d^ (*p* > 0.05) HbA_1c_ −26.7% for responder ^c^ (*p* < 0.001); NS decrease for non-responder ^d^ (*p* > 0.05) Insulin −52.6% for responder ^c^ (*p* < 0.001); −46.9% for non-responder ^d^ (*p* < 0.01) TAG −38.9% for responder ^c^ (*p* < 0.001); −31.6% for non-responder ^d^ (*p* < 0.05) HDL-C +27.3% for responder ^c^ (*p* < 0.01); +20.0% for non-responder ^d^ (*p* < 0.05)
(c) Overfeeding						
LIPOGAIN-Trial Rosqvist et al. (2014) [71]	7 wk n = 37 (26 M, 11 F) 20 to 38 y non-T2D non-OB (BMI: 18 to 27 kg/m^2^)	Two arms (RCT): Group 1: PUFA Group 2: SFA Muffin composition: 51 en% FAT, 5 en% PRO, 44 en% CHO individually adjusted to achieve 3% weight gain	1.5 Tesla MRI (Achieva, Philips, the Netherlands)	No increase and difference between PUFA vs. SFA at 7 wk (*p* > 0.05)	Increase at 7 wk: VAT Increase of +9.1% for PUFA vs. +25.0% for SFA (*p* < 0.05) PFC No increase and difference between PUFA vs. SFA (*p* > 0.05) LFC Increase of +0.0% for PUFA vs. +50.0% for SFA (*p* < 0.05)	No increase and difference between PUFA vs. SFA at 7 wk: FPG (*p* > 0.05) Insulin (*p* > 0.05) HOMA IR (*p* > 0.05)
LIPOGAIN2-Trial Rosqvist et al. (2019) [72]	12 wk n = 60 (37 M, 23 F) 20 to 55 y non-T2D OW (BMI: 25 to 32 kg/m^2^)	Two arms (RCT) Group 1: PUFA Group 2: SFA Muffin composition: 51 en% FAT, 5 en% PRO, 44 en% CHO aim 3% weight gain Then LED for 4 wk 800 kcal/day 18 en% FAT, 26 en% PRO, 52 en% CHO	1.5 Tesla MRI (Achieva, Philips, the Netherlands)	No increase and difference between PUFA vs. SFA at 12 wk (*p* > 0.05)	Change at 12 wk: VAT No difference between PUFA vs. SFA (*p* > 0.05) PFC No difference between PUFA vs. SFA (*p* > 0.05) LFC Decrease of −2.2% for PUFA vs. increase of +51.7% for SFA (*p* < 0.01)	No increase and difference between PUFA vs. SFA at 12 wk: FPG (*p* > 0.05) Insulin (*p* > 0.05) HOMA IR (*p* > 0.05)
(d) Snacking						
ATTIS-Trial Dikariyanto et al. (2020) [73]	8 wk n = 107 30 to 70 y non-T2D OW and OB (BMI: ≥23 kg/m^2^)	Two arms (RCT): Roasted almond, 20% of EER Control group (isoenergetic): Sweet and savoury mini-muffins, 20% of EER Consume between meals and avoid extra nuts/nut products	1.5 Tesla MRI (Magnetom Aera, Siemens, Erlangen, Germany) LFC quantified using HOROS V 1.1.7 software	No increase or difference in BMI in almond vs. control at 8 wk (*p* > 0.05)	No increase and difference between almond vs. control at 8 wk: VAT (*p* > 0.05) PFC (*p* > 0.05) LFC (*p* > 0.05)	No increase and difference between almond vs. control at 8 wk: FPG (*p* > 0.05) Insulin (*p* > 0.05) HOMA IR (*p* > 0.05) TAG (*p* > 0.05) HDL-C (*p* > 0.05) ALT (*p* > 0.05) Change at 8 wk for: Non-HDL-C Decrease of −2.8% for almond vs. +2.8% for control (*p* < 0.05) LDL-C Decrease of −2.4% for almond vs. +4.1% for control (*p* < 0.05)
(e) Isocaloric diet						
Guiseppe et al. (2018) [74]	8 wk n = 39 35 to 75 y T2D abdominal obese (M: >102 cm; F: >88 cm) HbA_1c_ ≤ 7.5% TAG ≤ 3.95 mmol/L LDL ≤ 3.36 mmol/L	Two arms (RCT): Multifactorial diet Higher fiber, polyphenols, PUFA and antioxidant MUFA diet Both diets are isoenergetic (~800 kcal) with 40%en CHO, 18%en PRO and 42%en FAT	3.0 Tesla MRI (dStream, Philips, the Netherlands) using a 2-point DIXON method with flexible echo times	Decrease at 8 wk: −1.5% for multifactorial diet vs. −1.2% for MUFA diet (*p* > 0.05)	Change at 8 wk: PFC −8.0% for multifactorial diet vs. 10.0% for MUFA diet (*p* < 0.05)	Change at 8 wk FPG 1.5% for multifactorial diet vs. 0.0% for MUFA diet (*p* > 0.05) Insulin −15.8% for multifactorial diet vs. 5.3% for MUFA diet (*p* > 0.05) HOMA IR −19.0% for multifactorial diet vs. 6.8% for MUFA diet (*p* > 0.05) HbA_1c_ −3.1% for multifactorial diet vs. −1.5% for MUFA diet (*p* > 0.05)
Physical activity intervention						
(a) Strength and endurance training						
Langleite et al. (2016) [16]	12 wk 40 to 65 y Sedentary Dysglycemic (n = 11 M) FPG ≥ 5.6 mmol/L OW (BMI: 27 to 32 kg/m^2^) Control (n = 11 M) FPG < 5.6 mmol/L lean (BMI: 19 to 25 kg/m^2^)	Two arms (non-RCT) 4 h wk intensive training, including 2 whole-body strength-training sessions and 2 spin bike interval sessions	1.5 T Tesla MRI (Achieva, Philips, the Netherlands) using 3D DIXON technique	NS decrease for dysglycemic at 12 wk (*p* > 0.05) NS decrease for control at 12 wk (*p* > 0.05)	Change at 12 wk: PFC NS decrease between dysglycemic vs. control at 12 wk (*p* > 0.05) LFC Decrease of −4.8% for dysglycemic vs. −1.0% for control at 12 wk (*p* < 0.05)	NS decrease between dysglycemic vs. control at 12 wk: FPG (*p* > 0.05) Insulin (*p* > 0.05)
(b) Short-term exercise training						
Heiskanen et al. (2018) [17]	2 wk Healthy (28 M) 40 to 55 y Sedentary lean and OW (BMI: 18.5 to 30 kg/m^2^) Pre-/T2D (16 M, 10 F) Sedentary lean, OW and OB (BMI: 18.5 to 35 kg/m^2^)	Two arms (RCT) Sprint interval training: 4 to 6 episodes of all-out cycling (30 s each) with supramaximal workload Moderate-intensity continuous training: 40–60 min cycling with 60% peak workload	1.5 Tesla MRI (Gyroscan Intera, Philips, the Netherlands)	NS decrease and difference between pre/T2D vs. healthy at 2 wk (*p* > 0.05)	NS decrease and difference between pre/T2D vs. healthy at 2 wk: VAT (*p* > 0.05) PFC (*p* > 0.05) LFC not assessed	No difference between pre/T2D vs. healthy at 2 wk: FPG (*p* > 0.05) HbA_1c_ (*p* > 0.05) Insulin (*p* > 0.05)
Combination of diet and physical activity intervention						
Vogt et al. (2016) [75]	15 wk n = 29 (10 M, 19 F) 18 to 70 y T2D OW and OW (BMI: ≥ 27 kg/m^2^)	Single arm (non-RCT) VLED Liquid diet (800 kcal) for 6 wk Refeeding phase (1200 kcal) for 4 wk Normal diet for 5 wk Cardio and strength training	3.0 Tesla MRI (Verio, Siemens, Erlangen, Germany)	Decrease at 6 wk −9.6%, *p* < 0.001 Decrease at 15 wk −12.7%, *p* < 0.001	Decrease at 6 wk: VAT (−18.8%, *p* < 0.001) LFC (−62.7%, *p* < 0.001) NS decrease at 6 wk for: PFC (*p* > 0.05) Decrease at 15 wk: VAT (−34.4%, *p* < 0.001) LFC (−71.1%, *p* < 0.001) NS decrease at 15 wk for: PFC (*p* > 0.05)	Decrease at 15 wk: TAG (−13.0%, *p* < 0.01) ALT (−33.1%, *p* < 0.001) AST (0.0%, *p* < 0.01) NS decrease at 15 wk for: HDL-C (*p* > 0.05)
CENTRAL-Trial Gepner et al. (2018) [76]	18 mo n = 278 11% F 28 to 69 y abdominal obese (M: >102 cm; F: >88 cm) dyslipidemia (>8.3 mmol/L) low HDL-C (M: <2.2 mmol/L; F: < 2.8 mmol/L)	Two arms (RCT) LF group 30 en% FAT, ≤ 10% SFA, ≤ 300 mg cholesterol/day MED/LC group CHO < 40 g/day in first 2 mo, then increase CHO ≤ 70 g/day and increase PRO and FAT intake 3 sessions per wk of aerobic and resistance training	3.0 Tesla MRI (Ingenia, Philips, the Netherlands) utilising modified 3D DIXON method	Decrease at 18 mo −3.2% for LF and MED/LC (*p* < 0.05)	Decrease at 18 mo: VAT −26.9% for LF/PA+ vs. −18.6% for LF/PA- (*p* < 0.05) −25.7% for MED/LC/PA+ vs. −19.3% for MED/LC/PA- (*p* < 0.05) NS difference LF vs. MED/LC, (*p* > 0.05) PFC −0.6% for LF/PA+ vs. −3.9% for MED/LC/PA+ (*p* < 0.05) Increase of +0.6% for LF/PA- vs. decrease of −2.3% for MED/LC/PA-(*p* < 0.05) NS difference PA+ vs. PA- (*p* > 0.05) LFC −42.3% for LF/PA+ vs. −44.8% for MED/LC/PA+ (*p* < 0.05) −34.3% for LF/PA- vs. −36.6% for MED/LC/PA- (*p* < 0.05) NS difference PA+ vs. PA- (*p* > 0.05)	Decrease at 18 mo: TAG −15.4% for MED/LC (*p* < 0.05) −4.8% for LF (*p* < 0.05) TAG/HDL Ratio −0.2 for MED/LC (*p* < 0.05) −0.1 for LF (*p* < 0.05) HDL-C −0.2 for MED/LC (*p* < 0.05) −0.3 for LF (*p* < 0.05)
Surgical intervention						
(a) Laparoscopic sleeve gastrectomy						
Umemura et al. (2017) [77]	6 mo n = 27 (14 M, 13 F) 18 to 65 y OB (BMI: ≥35 kg/m^2^)	Single arm (non-RCT)	VAT and PFC measured using a 64-row CT (Aquilion^TM^; Toshiba; Tokyo, Japan)	Decrease at 6 mo −27.5% (*p* < 0.001)	Decrease at 6 mo: VAT (−61.6%, *p* < 0.001) Increase at 6 mo for: Pancreatic attenuation ^e^ (+36.8%, *p* < 0.001)	Decrease at 6 mo: FPG (−28.1%, *p* < 0.01) HbA_1c_ (−35.8%, *p* < 0.01) HOMA IR (−71.3%, *p* < 0.001) TAG (−33.0%, *p* < 0.01) AST (−56.1%, *p* < 0.001) ALT (−75.0%, *p* < 0.001) Increase at 6 mo for: HDL-C (+11.1%, *p* < 0.05)
Covarrubias et al. (2019) [78]	6 mo n = 9 (2 M, 7 F) ≥18 y OB (BMI: ≥35 kg/m^2^)	Single arm (non-RCT)	3.0 Tesla MRI (GE Signa EXCITE HDxt, GE Healthcare, Waukesha, WI); 3D multi-echo spoiled gradient echo (SGRE) sequence	Decrease at 6 mo −26.0% ^f^	Decrease at 6 mo: VAT (−40.8%) ^f^ PFC (−39.9%, *p* < 0.01) LFC (−77.0%) ^f^	Not reported
(b) Roux-en-Y gastric bypass						
Steven et al. (2016) [79]	8 wk Group 1 n = 18 (7 M, 11 F) T2D < 15 y Group 2 n = 9 (2 M, 7 F) NGT Both groups 25 to 65 y BMI: ≤ 45 kg/m^2^	Two arms (non-RCT)	3.0 Tesla MRI (Philips, the Netherlands) using a 3-point Dixon method	Decrease at 8 wk −13.7% for T2D (*p* < 0.001) −12.9% for NGT (*p* < 0.001)	Decrease at 8 wk: VAT −19.7% for T2D (*p* < 0.001); −23.1% for NGT (*p* < 0.05) PFC −18.2% for T2D (*p* < 0.01); NS decrease for NGT (*p* > 0.05) LFC −44.1% for T2D (*p* < 0.05); NS decrease for NGT (*p* > 0.05)	Decrease at 8 wk: FPG −32.6% for T2D (*p* < 0.001); NS decrease for NGT (*p* > 0.05) Insulin −26.1% for T2D (*p* < 0.001); 39.1% for NGT (*p* < 0.01) TAG −26.7% for T2D (*p* < 0.05); NS decrease for NGT (*p* > 0.05) ALT −31.8% for T2D (*p* < 0.01); NS decrease for NGT (*p* > 0.05)
Lautenbach et al. (2018) [80]	6 mo n = 11 ≥18 y OB (BMI: ≥30 to ≤50 kg/m^2^)	Single arm (non-RCT)	3.0 Tesla MRI (Ingenia, Philips, Germany)	Decrease at 6 mo −24.6%, *p* < 0.001	Decrease at 6 mo: VAT (−48.3%, *p* < 0.001) PFC (−45.5%, *p* < 0.01)	Decrease at 6 mo: HbA_1c_ (−13.5%, *p* < 0.01) Insulin (−64.9%, *p* < 0.01) HOMA IR (−67.5%, *p* < 0.01) ALT (−28.8%, *p* < 0.05) NS decrease at 6 mo for: FPG (*p* > 0.05) TAG (*p* > 0.05) HDL-C (*p* > 0.05)
(c) Comparison of surgical methods						
Gaborit et al. (2015) [81]	6 mo n = 20 (8 T2D, 12 non-T2D) 43.3 ± 1.8 y (6 M, 14 F) Severely OB (BMI: > 40 or ≥ 35 kg/m^2^ with at least one comorbidity)	Two arms (non-RCT) with both LSG and RYGB	3.0 Tesla MRI (Verio, Siemens, Erlangen, Germany)	Decrease at 6 mo −24.3% for T2D and NGT (*p* < 0.001)	Decrease at 6 mo:VAT −40.6% for T2D (*p* < 0.05); −46.3% for NGT (*p* < 0.01) PFC −52.9% for T2D and NGT (*p* < 0.001) LFC −69.7% for T2D and NGT (*p* < 0.001)	Decrease at 6 mo: FPG −32.4% for T2D (*p* < 0.05); −13.7% for NGT (*p* < 0.01) HbA_1c_ −26.3% for T2D (*p* < 0.05); NS decrease for NGT (*p* > 0.05) Insulin NS decrease for T2D (*p* > 0.05); −70.8% for NGT (*p* < 0.001) TAG NS decrease for T2D (*p* > 0.05); −15.4% for NGT (*p* < 0.05) HDL-C NS decrease for T2D and NGT (*p* > 0.05) AST NS decrease for T2D and NGT (*p* > 0.05) ALT −38.8% for T2D (*p* < 0.05); NS decrease for NGT (*p* > 0.05)
Honka et al. (2015) [82]	6 mo n = 23 (10 T2D, 13 non-T2D) 18 to 60 y Severely OB (BMI: > 40 or ≥ 35 with additional risk factor)	Two arms (non-RCT) with both LSG and RYGB	1.5 Tesla MRI (Gyroscan Intera CV Nova Dual, Philips, the Netherlands)	NS decrease and difference between T2D vs. NGT at 6 mo (*p* > 0.05)	Decrease at 6 mo: VAT No difference between T2D vs. NGT (*p* > 0.05) PFC −30.0% for both (*p* < 0.01)	Decrease at 6 mo: TAG Decrease of −6.7% for T2D vs. −10.0% for NGT (*p* < 0.05) HDL-C Increase of −13.2% for T2D vs. −17.0% for NGT (*p* < 0.05) NS decrease and difference between T2D vs. NGT at 6 mo for: FPG (*p* > 0.05) HbA_1c_ (*p* > 0.05) Insulin (*p* > 0.05) HOMA IR (*p* > 0.05)
Hui et al. (2019) [83]	12 mo n = 12 (4 M, 8 F) 18 to 65 y OB (BMI: ≥ 35 or ≥ 30 with MetS)	Single arm (non-RCT) with LSG, RYGB and LGCP	3.0 Tesla MRI (Achieva, Philips, the Netherlands)	Decrease at 12 mo −29.3%, *p* < 0.001	Decrease at 12 mo: VAT (−51.2%, *p* < 0.001) PFC (−67.2%, *p* < 0.05) LFC (−93.9%, *p* < 0.001)	Not reported
Comparison of diet and surgical intervention						
Steven et al. (2016) [84]	1 wk n = 18 (9 RYGB, 9 VLED) 25 to 65 y T2D Severely OB (BMI: ≤ 45 kg/m^2^)	Two arms (RCT) RYGB VLED 700 kcal/day	3.0 Tesla MRI (Philips, the Netherlands)	Decrease at 1 wk −5.1% for RYGB vs. −3.5% for VLED (*p* < 0.05)	Decrease at 1 wk: PFC NS decrease for RYGB or VLED (*p* > 0.05) LFC Decrease of −29.8% for RYGB (*p* < 0.05) Decrease of −18.6% for VLED (*p* < 0.01) NS difference between RYGB vs. VLED (*p* > 0.05)	Decrease at 1 wk: FPG NS decrease for RYGB and VLED (*p* > 0.05) Insulin NS decrease for RYGB and VLED (*p* > 0.05) TAG NS decrease for RYGB and VLED (*p* > 0.05) ALT Increase of +15.0% for VLED (*p* < 0.05); NS decrease for RYGB (*p* > 0.05)

^a^ Responder: FPG < 7.0 mmol/L after return to an isoenergetic diet; ^b^ Non-responder: FPG ≥ 7.0 mmol/L after return to an isocaloric diet; ^c^ Responder: Hb_A1c_ < 6.5% and FPG < 7.0 mmol/L with no glucose-suppressing medications; ^d^ Non-responder: Hb_A1c_ ≥ 6.5% and FPG ≥ 7.0 mmol/L with no glucose-suppressing medications; ^e^ Pancreatic attenuation: A measurement in 3 regions of interest in pancreas and spleen on non-enhanced CT images, which had a significant negative correlation with pancreatic fat fraction as reported by Kim and colleagues (2014); ^f^
*p* value not reported in publication; Abbreviations: ALT: Alanine aminotransferase; AST: Aspartate aminotransferase; BMI: Body mass index; BW: Body weight; CCR: Continuous calorie restriction; CD: Conventional diabetes diet; CHO: Carbohydrates content; CRHP: Carbohydrate reduced high-protein diet; CT: Computed tomography; en%: percentage of energy; F: Female; FAT: Fat content; FPG: Fasting plasma glucose; HDL-C: High-density lipoprotein cholesterol; HOMA-IR: Homeostatic model assessment of insulin resistance; ICR: Intermittent calorie restriction; LED: Low-energy diet; LF: Low fat; LFC: Liver fat content; LGCP: Laparoscopic greater curvature plication; LSG: Laparoscopic sleeve gastrectomy; M: Male; MED/LC: Mediterranean low-carbohydrate diet; MetS: Metabolic syndrome; mo: month;, MRI: Magnetic resonance imaging; MRS: Magnetic resonance spectroscopy; NS, no significant; OB: obese; OW: overweight; PA: Physical activity; PAL: Physical activity level; PFC: Pancreatic fat content; PRO: Protein content; PUFA: Polyunsaturated fatty acids; RCT: Randomised controlled trial; REE: Resting energy expenditure; RYGB: Roux-en-Y gastric bypass; SFA: Saturated fatty acids: T2D: Type 2 diabetes; TAG: Triglycerides; VAT: Visceral adipose tissue; VLED: Very low-energy diet; WC: waist circumference; wk: week; y: year.

Meanwhile, it is important to emphasise that individuals in both groups were required to lose ~3.8% of their baseline body weight during week 1 of the VLED to continue and progress through to the full study. Both groups also achieved a significant decrease in VAT (~30%), PFC (~10 to 17%) and LFC (~70 to 80%) after 8 weeks of VLED intervention. Notably, the decrease was maintained only for VAT (~30%), but not for either PFC or LFC at 6 months. In terms of metabolic parameters at 8 weeks, there were significant improvements reported for fasting insulin (~41 to 61%) and TAG (~23 to 35%) for both groups whereas only responders had a significant improvement in their alanine aminotransferase (ALT) level (~40%). There was no significant improvement in HDL-C levels for both groups. After 6 months of intervention, the improvement in fasting insulin (~37 to 63%) and TAG (~8 to 40%) remained significant for both groups. Surprisingly, non-responders reported a ~4% decrease in ALT after 6 months while responders had a further ~11% reduction. In terms of HDL-C, responders achieved a significant increase of ~27%, indicating improved protection against cardiometabolic disease, but not among the non-responders.

In contrast, the recent T2D intervention study DiRECT [70] with a longer 3 to 5-month VLED intervention and a follow-up period of 24 months, reported a significant decrease in body weight for both responders (HbA_1c:_ < 6.5%, FPG: < 7 mmol/L, no medications) and non-responders after 5 months (responder: ~16%; non-responder: ~13%) as well as after 24 months (both groups: ~12%) following intervention with the Counterweight-Plus weight management program. Again, this was a single treatment study, assessing change from baseline. In addition, VAT, PFC, and LFC all decreased significantly regardless of whether the individuals with T2D achieved remission or not after 3 to 5 months of VLED intervention (VAT: ~36 to 43%; PFC: ~10%; LFC: ~80 to 82%). At the 24-month follow-up, the responders maintained a significant decrease in VAT (~27%), PFC (~8%), and LFC (~61%), albeit with some rebound in all three parameters, whereas non-responders maintained only VAT losses (~28%). It was notable that responders who maintained significant improvements in FPG and HbA_1c_ also had significant improvements in their fasting insulin, TAG, and HDL-C. Non-responders, in whom FPG and HbA_1c_ had not normalised during the 3 to 5-month VLED, unexpectedly, however, did report improvements in fasting insulin, as well as in TAG and HDL-C after 24 months.

### 4.3. Overfeeding Intervention Trials which Include Snacking

These interventions required the individuals to consume snacks as part of their habitual diet, with the intent to compare the effect of overconsumption of different types of dietary fat on ectopic fat accumulation and metabolic parameters. Two such weight gain trials have been conducted by Rosqvist and colleagues, LIPOGAIN and LIPOGAIN 2, both of which manipulated the dietary fat composition of the snacks given to the individuals. The first trial [71] provided high polyunsaturated fatty acid (PUFA) or high saturated fatty acid (SFA) muffins to healthy individuals over 7 weeks. Both groups achieved an additional 3% body weight gain, as required by the protocol, but the PUFA group reported a lower VAT increment (PUFA: 9.1% vs. SFA: 25.0%) and no increase in LFC (PUFA: 0.0% vs. SFA: 50.0%) compared to the SFA group. Meanwhile, there was no significant difference in change for PFC, FPG, insulin, or homeostatic model assessment for insulin resistance (HOMA-IR) for either group. In the latter 12-week LIPOGAIN 2 study [72], the authors prolonged the PUFA vs. SFA overfeeding/weight gain period to 8 weeks followed by 4 weeks of VLED weight loss. There was no difference in weight gain/loss between the two groups, but LFC decreased by 2.2% in the PUFA group whereas it increased by 51.7% in the SFA group. There was no significant difference in either VAT or PFC. Similar results were reported for the metabolic endpoints, such as FPG, insulin, and HOMA-IR for both groups.

On the other hand, the ATTIS (Almonds Trial Targeting Dietary Intervention with Snacks) study [73] applied a similar approach of manipulating dietary fat composition, but with no intent of overfeeding or weight gain, by providing whole roasted almonds or sweet and savoury muffins (control) as a snack for 8 weeks. Whilst body weight was not reported, BMI did not change significantly in either group. There was also no significant change in VAT, PFC, and LFC nor in metabolic parameters of FPG, insulin, HOMA-IR, TAG, HDL-C, or ALT in the almond or control groups over 8 weeks, nor any difference between the treatment groups. Whilst glycaemic endpoints were not significantly altered, cardiovascular disease (CVD) risk markers of non-HDL-C, and low-density lipoprotein cholesterol (LDL-C) were significantly decreased in the almond vs. control group.

### 4.4. Isocaloric Diet Intervention

The study was conducted by Giuseppe and colleagues [74] among the T2D patients, and they are being randomly assigned to either a multifactorial diet or a monounsaturated fatty acid (MUFA) diet for 8 weeks. Both diets are isoenergetic and a similar macronutrient composition, but the multifactorial diet contains more fibre, polyphenols, PUFA, and antioxidants. After 8 weeks, there were no significant differences between the groups in terms of body weight loss (Multifactorial: −1.5% vs. MUFA: −1.2%, *p* > 0.05), FPG (Multifactorial: 1.5% vs. MUFA: 0.0%, *p* > 0.05), fasting insulin (Multifactorial: −15.8% vs. MUFA: 5.3%, *p* > 0.05), HOMA-IR (Multifactorial: −19.0% vs. MUFA: 6.8%, *p* > 0.05), and HbA_1c_ (Multifactorial: −3.1% vs. MUFA: −1.5%, *p* > 0.05). However, a significant difference was reported on the changes in PFC between the groups (Multifactorial: −8.0% vs. MUFA: 10.0%, *p* > 0.05).

### 4.5. Physical Activity Interventions

There are two studies that have utilised exercise regimes to determine their impact on body fat compartments, especially pancreatic and liver fat, as well as metabolic parameters. The first trial [16] assigned overweight dysglycaemic and lean healthy individuals to undergo a 12-week training regime that focused on strength and endurance. Individuals with dysglycaemia had a significantly greater decrease in LFC compared to healthy individuals (dysglycaemic: −4.8% vs. healthy: −1.0%, *p* < 0.05) after 12 weeks, despite no change in body weight for either group. Otherwise, there was no significant change in PFC, FPG, or fasting insulin, regardless of health status.

The second trial [17] utilised a shorter 2-week training regime that focused on aerobic fitness in lean and overweight individuals with normoglycaemia or prediabetes. LFC was not assessed in this trial however and so the observations of the first trial [16] cannot be compared. Again, there was no significant decrease in the baseline or difference between participant groups in body weight, VAT, PFC, FPG, or fasting insulin after the very short intervention of 2 weeks of aerobic training.

### 4.6. Combined Dietary and Physical Activity Interventions

The first trial in this category [75] utilised a combination of VLED liquid formula diet (6 weeks) and then re-feeding (4 weeks) followed by a normal diet alongside a training regime (5 weeks) focused on cardio and strength exercises in individuals with obesity and T2D. This was a single-arm, non-randomised trial that assessed change from baseline, with no comparison group. After 6 weeks, there was a significant decrease of ~7% in body weight (*p* < 0.001), ~19% in VAT (*p* < 0.001) and ~63% in LFC (*p* < 0.001), but despite these improvements, there was no significant change in PFC. The significant decreases in body weight (~13%), VAT (~34%), and LFC (~71%) were maintained at the end of the 15-week trial with a significant decrease in fasting TAG, ALT, and aspartate aminotransferase (AST), but no change in HDL-C.

A much longer 18-month RCT, the CENTRAL-Trial [76], utilised either a low-fat diet (LF) or Mediterranean/low-carbohydrate diet (MED/LC), with or without a physical activity (PA) training regime focusing on aerobic and resistance exercise among individuals with abdominal obesity and dyslipidaemia. All intervention arms resulted in a decrease of −3.2% body weight (*p* < 0.05) regardless of the type of diet and training regime. Meanwhile, individuals who underwent the exercise training regime (PA+) lost more VAT compared to those without the training regime (PA-) regardless of the diet. Conversely, individuals randomised to the MED/LC diet had a greater decrease in PFC (PA+: −3.9% vs. −0.6%; PA-: −2.3% vs. +0.6%) and LFC (PA+: −44.8% vs. −42.3%; PA-: −36.6% vs. −34.3%), but not VAT compared to those with LF diet regardless of their training regime. TAG was also decreased in MED/LC (~15%) vs. LF (~5%) after 18 months.

### 4.7. Surgical Interventions

There are seven studies that utilised several common bariatric surgical procedures, such as laparoscopic sleeve gastrectomy (LSG), Roux-en-Y gastric bypass (RYGB), and laparoscopic greater curvature plication (LGCP). RYGB is known as the ‘gold standard’ among surgical procedures, where a small pouch is created from the stomach and connected to the bottom end of the divided small intestine to create a new route for the digestion of food. Meanwhile, LSG is a similar surgical procedure to RYGB in that it removes ~80% of the stomach and directly connects to the small intestine without resecting that region [85]. As both RYGB and LSG require the irreversible reduction in gastric volume by removal of most of the stomach, LGCP is a newer minimally invasive surgical procedure with many similarities to LSG but involves multiple surgical sutures to create a smaller stomach volume instead of partial resection of the stomach [86]. These bariatric procedures restrict the amount of food consumed to promote rapid and significant weight loss and T2D remission, especially in individuals with significant obesity [77,78,79,80,81,82,83].

A significant decrease in body weight among individuals who underwent these surgical procedures, including LSG (~28%), RYGB (~13 to 25%), and a combination of different procedures (~24 to 29%) was observed. In addition, LSG also decreased VAT by ~62%, as did the combination of procedures (~41 to 51%) and RYGB (~20 to 48%). In terms of PFC, the key outcome of this review, LSG resulted in a decrease of ~40%, as well as improved pancreatic attenuation of ~37%, which implies a lower pancreatic fat fraction by measuring the three regions of interest in the pancreas on non-enhanced CT images [87]. Individuals with RYGB or a combination of surgeries decreased PFC by ~18 to 46% and ~30 to 67%, respectively. However, unexpectedly, a small cohort of normoglycaemic individuals with severe obesity undergoing RYGB intervention failed to decrease PFC despite almost 13% acute body weight loss over 8 weeks [84]. Three of seven studies also reported LFC data whereby RYGB decreased fatty liver by ~44% and combination of surgical procedures by ~70 to 94%. In summary, the combination of surgical procedures achieved the greatest decrease in body weight (~29%), PFC (~67%), and LFC (~94%), whereas LSG achieved the greatest decrease in VAT (~62%). However, the results of the three trials that utilised the combination of surgical approaches should be interpreted carefully as the author’s reported outcomes were focused on the combined overall metabolic changes following surgical procedures rather than the individual effect following each of these surgical procedures [81,82,83].

One study reported a comparison of dietary and surgical outcomes and compared VLED with RYGB surgical intervention for weight loss, albeit in a very short study of only 1-week duration. Steven and colleagues [84] reported a significant decrease in body weight below the pre-intervention baseline (Diet: −3.5% vs. Surgery: −5.1%) despite the short duration. There was, however, no significant effect of either intervention on PFC, although LFC was significantly decreased over the 7-day study period (Diet: −18.6%, *p* < 0.01; Surgery: −29.8%, *p* < 0.05).

## 5. Discussion—Can We Identify the Best Practice Approach for the Amelioration of Ectopic Fat and Cardiometabolic Risk?

Of the 22 trials reported in this narrative review, 13 studies reported a significant decrease in body weight following the adherence to a weight-loss strategy [15,62] comprising VLED total meal replacement [68,69,70], isocaloric diet [74], the combination of diet and physical activity [75,76], surgical procedures [77,79,80,83], as well as a comparison between VLED and RYGB [84]. Among the 13 clinical studies, most of the trials reported a significant decrease in VAT, PFC, and LFC. It is important to note that, while all reported a significant decrease in both VAT and LFC, this was not observed for PFC in three [15,75,84] of the 13 trials. An evaluation of the MR methodologies in these three clinical studies revealed different techniques used to measure PFC, and the results showed no consensus on the association between the changes in PFC and T2D. One possible explanation for the lack of change in PFC in the three trials could be due to the shorter intervention duration. Two studies [75,84] used short-duration VLEDs of 1 week and 6 weeks, respectively, in comparison to the other studies reported in this review where VLED was implemented for at least 8 weeks. It, therefore, appears likely that a minimum of 8 weeks may be required to drive significant changes in PFC and improvements in glycaemic outcomes. This warrants further investigation in controlled clinical studies that report PFC after 8+ weeks of VLED-weight loss to unravel and better understand these relationships.

Among the 13 trials that achieved significant body weight loss, surgical intervention achieved the largest decrease in PFC (range: −18.2% to 67.2%) compared to either dietary intervention (weight loss diet, isocaloric diet, and/or VLED, range: −10.2% to −42.3%) or combination of diet and physical activity (range: −0.6% to 3.9%). The rapid impact of bariatric surgery on the glycaemic status and, in turn, consequent T2D reversal may perhaps be driven through a rapid decrease in PFC. This has been proposed due to changes in gastrointestinal hormone secretion that regulates adiposity [88] or alteration in fatty acid transport proteins that lower fatty acid update in the pancreas [89], amongst other potential mechanisms. As a result, PFC loss may directly promote better insulin response and β-cell function, which is beneficial towards the rapid remission of T2D [82].

Meanwhile, the small change in PFC following the combination of diet and physical activity, while unexpected, may likely be attributed to the diets used, where only macronutrient composition was manipulated without a change in total energy. This approach contrasts with the much greater change in PFC following energy-restricted diets such as the VLEDs. Meanwhile, overfeeding, snacking, and physical activity interventions were unable to trigger any significant changes in body weight, body fat compartments, or metabolic parameters, except for LFC. There were studies hypothesising that LFC may be mobilised more rapidly within a shorter duration of energy restriction, which is around 2 to 8 weeks, compared to visceral and subcutaneous fat [62,90]. This hypothesis is supported by our current results with some evidence that a minimum of 8 weeks of VLED or surgical intervention might be required to observe significant changes in PFC. PFC was markedly decreased following VLED intervention albeit some rebound was reported during the follow-up or maintenance periods.

In addition, decreased body weight and body fat compartments were accompanied by significant decreases in key metabolic parameters, including fasting glucose, insulin, HbA_1c_, and TAG. There was a significant decrease in both fasting glucose and TAG especially in VLED trials [68,69,70], the combination of diet and physical activity [75,76], LSG [77], and RYGB surgery [79]. Overall, the evidence from the clinical intervention studies of between 2 to 24 months duration among individuals with overweight and obesity, as well as variable glycaemic and diabetic status, supports the proposal that a successful decrease in total body weight and adiposity achieved through lifestyle intervention or bariatric surgery may drive a significant decrease in PFC and LFC, and, in turn, improvement in cardiometabolic risk. However, there is no evidence for decreases in ectopic fat, including PFC, following exercise interventions in the absence of weight loss.

Although studies utilising bariatric surgery reported a greater and significant decrease in pancreatic and liver fat in comparison to studies that utilised dietary intervention, surgery is invasive and not all individuals have equitable access. Therefore, dietary modification using weight loss diets over the longer-term duration of 12+ weeks or VLEDs over the short/medium-term duration of 8+ weeks are likely to be the best alternatives for the reduction in ectopic fat, in comparison to body weight and total adipose mass loss for individuals at risk of cardiometabolic disease. However, it is crucial to emphasize that higher body weight loss does not always guarantee better efficacy in pancreatic fat loss. Based on the results of this review, individuals who followed a standard weight loss diet regime lost less body weight than those who followed the ≤4 MJ VLED regime, with the latter driving a greater negative energy balance, but both groups lost similar PFC. We can, therefore, hypothesise that pancreatic fat may be a rapid, early depot from which lipid is lost under the conditions of energy restriction and/or that PFC may be mediated by other additional factors, not just solely depending on body weight change.

Dietary modification is the cornerstone in the management and prevention of adverse metabolic health and T2D. There is consensus for formulating evidence-based dietary guidelines, which have evolved from low-fat diets towards wider consideration of macronutrient quality and quantity with a focus on longer-term weight management and prevention of weight regain through the maintenance of long-term energy balance. Recent evidence from large intervention trials such as the DiRECT [70] and the DROPLET [64] studies in the United Kingdom have provided robust evidence that VLEDs used as part of a long-term weight-loss maintenance programme can benefit individuals with overweight or obesity and achieve remission of T2D. The deficit in energy intake results in the utilisation of body fuel reserves from adipose tissue stores, including within the liver, which in turn can lead to the normalisation of insulin sensitivity with a concomitant decrease in pancreatic fat and improvement in pancreatic beta-cell function via re-differentiation following fat loss [91]. VLED and meal replacements, therefore, may provide an alternative non-invasive strategy, at a lower cost to both individuals and government health programmes than bariatric surgery. Clearly, there is a need to focus strategies of long-term maintenance after weight loss and the prevention of weight regain, using VLEDs, with possible scope for low- to very low (based on ≤ 8MJ/day)-carbohydrate diet interventions as discussed by Taylor and colleagues [92], alongside low-fat and higher-protein [67] strategies. Currently, there is little data published from long-term follow-up studies assessing PFC, but that may be of interest going forwards for future research.

## Figures and Tables

**Table 1 nutrients-14-04873-t001:** Techniques for measurement of pancreatic and liver fat.

Technique	Advantages	Disadvantages
Ultrasound	Non-invasive Fast procedure No exposure to radiation Economic	Operator dependent Low reproducibility Low accuracy due to indirect measurement
CT ^1^	Good accessibility Reproducible High specificity High accuracy	Exposure to radiation Expensive Weight limitation
MRI ^2^/MRS ^3^	Best spatial resolution and body mass composition differentiation Non-invasive No exposure to radiation High accuracy	Difficult to access the machine Expensive Time-consuming

^1^ CT: computed tomography; ^2^ MRI: magnetic resonance imaging; ^3^ MRS: magnetic resonance spectroscopy.

## Data Availability

Data supporting reported results can be found in the cited journal articles throughout the review.

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
