# Peer review of "Fatty Pancreas and Cardiometabolic Risk: Response of Ectopic Fat to Lifestyle and Surgical Interventions"

_nutrients, 2022, doi:10.3390/nu14224873_

Round 1
Reviewer 1 Report
This is a timely review of an area of metabolic disease which has been understudied considering the importance of the pancreas in metabolic disease and the known impacts of ectopic fat content in the liver.
The review is well structured, and the key information extracted and presented from relevant research studies in the field.
Minor comments:
1. The first paragraph of the section ‘Adverse metabolic effects attributed to accumulation of pancreatic fat’ is perplexing. I find no mention of ‘inter-pancreatic fat’ in Reference 31. Indeed there is very little mention of it on the internet. However, I do think that a description of the known causes for increased fat content in the pancreas (i.e. uptake or generation of lipids within the pancreas, versus adipocyte infiltration into the pancreas) is needed so this section should also be expanded.
2. Regarding the uncertainty of the causal nature of pancreatic fat content and T2D (described in P3-4) there will likely be some information on this issue in animal models. If so, please describe that literature.
3. Table 1 needs lines to separate the 3 sections. Without them it is unclear which advantages apply to ultrasound, CT or MRI. Similarly, lines are needed in Table 2.
4. A surprising outcome from bariatric surgery in patients with T2D is that the diabetes is rapidly corrected after surgery, well in advance of body weight loss (and presumably pancreatic fat reduction). Do the authors think this should be commented on in the review? It seems that this mystery could inform research that aims to understand the mechanisms between T2D and pancreatic fat content.
Author Response
We would like to express our gratitude to the reviewer for providing us with insightful comments, and for your efforts towards improving our manuscript. Please see the attachment as we have addressed and provided detailed responses to your comments.

Reviewer 2 Report
Thank you for the opportunity to review this paper. The aim of the study is to summarise the existing evidence on effects of lifestyle and surgical intervention on pancreatic fat content, and explore what intervention might be best for reducing pancreatic fat content. The aim is interesting as it focus on an important fat component-pancreas fat which is less studies but plays a vital role in modulating pancreas function in regulating glucose homeostasis and other metabolic outcomes. The authors provided concise description of the existing literature/evidence but the organization of data in Table 2 can be improved. There are some other studies e.g. Lilac Tene et al., 2018 and Giuseppe et al., 2022 which are directly related to the context of this manuscript that the authors should consider including in this review. The authors should provide a clearer conclusion. There are minor grammatical errors throughout the manuscript.
Abstract:
Lines 15-16: The authors stated that “evaluated relationships between intra-organ fat and metabolic outcomes” but no data/results/findings related to this evaluation was reported in the abstract.
Lines 19-21: Is there a mean/median value (apart from range) for the decrease in pancreatic fat for each type of intervention?
Lines 21-24: The conclusion does not seem to fit with the overall aim of the study which is to examine change in ectopic fat accumulation in pancreas (pancreas fat) in each intervention. The conclusion should be written with respect to which intervention is effective for reducing pancreas fats and thus reducing cardiometabolic risk. Please rephrase and revise.
Introduction:
Lines 28-29: The authors stated that “Ectopic fat infiltration occurs in non-adipose organs such as muscle, heart, liver, and pancreas, which are critical to the regulation of cardiometabolic health”, and the phrasing seems to imply that ectopic fat infiltration is beneficial for cardiometabolic health (which is not the case). Please kindly rephrase and revise.
Line 31: The authors did not provide description/explanation on what is or what caused lipotoxicity.
Lines 39-40: Ectopic fat accumulation or lipotoxicity is linked not only to insulin resistance but metabolic syndrome as well. Since the title of the manuscript is on fatty pancreas and cardiometabolic risk, the authors should add in more published findings on ectopic fat accumulation and metabolic syndrome/cardiometabolic risk.
Line 68: It is unclear how the structure of pancreas play a role in development of T2D. It is still the function of the pancreatic parenchyma that contributes to T2D and not the structure itself.
Line 135: Please insert “,” after “overweight”
Table 1: Please insert row lines to separate each technique for easy referencing. Please indicate clearly what “low accuracy” and “poor accessibility” mean?
Lines 154-155: The authors stated that “Despite these limitations, evidence underpinning the association between pancreatic fat and cardiometabolic endpoints is growing”. However, it is to note that authors have presented data/results that are limited to T2D and no other cardiometabolic end points (lines 124-142).
Table 2 is confusing. The authors should separate into multiple tables, each explaining the different type of intervention as indicated in the subsections e.g. weight-loss diet interventions, VLED total meal replacement interventions. Please remove the “Decrease at ……” for all sections in the table as the negative sign of values would already indicated the decrease. The positioning of Table 2 in the manuscript is inappropriate as it cuts off the text abruptly.
Lines 170-171: Please change to “body fat compartments including pancreatic fat and metabolic parameters”.
Line 187: What do the authors meant by “Despite the positive outcomes in the CCR group”, how about the ICR group? There are also positive outcomes (decrease in body weight and VAT) in the ICR group.
Line 203: At the start of this section, the authors should explain the difference between VLED total meal replacement intervention and weight loss diet intervention. This is because both seems to be weight-loss interventions so it will be good to explain why you have chosen to separate them.
Lines 225-226: Were the metabolic parameters available/reported after 8 weeks of VLED intervention?
Line 253: There should be no spacing in front as this is a continued sentence, please revise the positioning of Table 2 accordingly.
Line 279: Please remove “As previously”.
Line 351: Please remove “In turn”.
Line 365-366: Please also cite first author name followed by et al.
Lines 406-430: The authors seem to think that VLED intervention is the best dietary intervention for reducing pancreatic fat content. How about the weight loss intervention with calorie deficit? They seem to work as well as VLED intervention.
Lines 436-442: The authors should provide a clearer conclusion e.g. interventions that achieved higher weight loss will be more effective in reducing pancreatic fat content and longer term (>8weeks) calorie deficit diet may be a useful dietary approach.
Author Response

(The authors gave the same response as above.)

Round 2
Reviewer 2 Report
Thank you for the opportunity to review the revised manuscript. The authors have made substantial changes to the manuscript and responded clearly to the comments by the reviewer. The manuscript has improved with the revisions. This revised manuscript still lacks emphasis on other cardiometabolic risk factors as the authors mainly focused on insulin resistance and T2D. The authors attempted to include some information on association/relationship between pancreatic fat and other cardiometabolic risk factors e.g. lipid levels. However, these references (references 60 and 61) are systematic/meta-analysis review so the elaboration on the data is brief (data in other studies have been summarised in the cited reviews). The authors should note that it is best to cite the original research article instead of review when the nature of this manuscript is a narrative review.